# A Postmortem MRI Study of Cerebrovascular Disease and Iron Content at End-Stage of Fragile X-Associated Tremor/Ataxia Syndrome

**DOI:** 10.3390/cells12141898

**Published:** 2023-07-20

**Authors:** Jun Yi Wang, Gerard J. Sonico, Maria Jimena Salcedo-Arellano, Randi J. Hagerman, Veronica Martinez-Cerdeno

**Affiliations:** 1Center for Mind and Brain, University of California Davis, Davis, CA 95618, USA; 2Imaging Research Center, University of California Davis, Sacramento, CA 95817, USA; gjsonico@ucdavis.edu; 3Department of Pathology and Laboratory Medicine, University of California Davis School of Medicine, Sacramento, CA 95817, USA; mjsalcedo@ucdavis.edu; 4MIND Institute, University of California Davis Health, Sacramento, CA 95817, USA; rjhagerman@ucdavis.edu; 5Institute for Pediatric Regenerative Medicine and Shriners Hospitals for Children Northern California, Sacramento, CA 95817, USA; 6Department of Pediatrics, University of California Davis School of Medicine, Sacramento, CA 95817, USA

**Keywords:** FXTAS, fragile X premutation, *FMR1*, repeat expansion disorder, movement disorder, cerebrovascular disease, neurodegeneration, brain iron accumulation, aging

## Abstract

Brain changes at the end-stage of fragile X-associated tremor/ataxia syndrome (FXTAS) are largely unknown due to mobility impairment. We conducted a postmortem MRI study of FXTAS to quantify cerebrovascular disease, brain atrophy and iron content, and examined their relationships using principal component analysis (PCA). Intracranial hemorrhage (ICH) was observed in 4/17 FXTAS cases, among which one was confirmed by histologic staining. Compared with seven control brains, FXTAS cases showed higher ratings of T2-hyperintensities (indicating cerebral small vessel disease) in the cerebellum, globus pallidus and frontoparietal white matter, and significant atrophy in the cerebellar white matter, red nucleus and dentate nucleus. PCA of FXTAS cases revealed negative associations of T2-hyperintensity ratings with anatomic volumes and iron content in the white matter, hippocampus and amygdala, that were independent from a highly correlated number of regions with ICH and iron content in subcortical nuclei. Post-hoc analysis confirmed PCA findings and further revealed increased iron content in the white matter, hippocampus and amygdala in FXTAS cases compared to controls, after adjusting for T2-hyperintensity ratings. These findings indicate that both ischemic and hemorrhagic brain damage may occur in FXTAS, with the former being marked by demyelination/iron depletion and atrophy, and the latter by ICH and iron accumulation in basal ganglia.

## 1. Introduction

Short tandem repeats are multiple copies of DNA sequences, typically 1–6 nucleotides, that lie adjacent to each other without gaps. At least 6.77% of the human genome is short tandem repeats that are prone to mutate and become a source of genetic variation in human populations [1]. Large expansions of these repeat tracts, however, can cause neurological or developmental disorders presenting shared clinical phenotypes, including cerebellar ataxia, tremor, cognitive impairment and peripheral neuropathy [2]. Currently, over fifty repeat expansion disorders have been identified [3]. Fragile X-associated tremor/ataxia syndrome (FXTAS) is one of these disorders, due to the expansion of the CGG repeat element at a noncoding region of the fragile X messenger ribonucleoprotein 1 (*FMR1*) gene in the premutation range (55–200 repeats) [4,5].

FXTAS is an age-related neurodegenerative disorder affecting premutation carriers with a prevalence in males increasing from 17% in those in their 50’s, to 47% in those in their 70’s, and to 75% in those aged 80 and above [6]. In female premutation carriers, FXTAS prevalence is lower but also increases with age, from 3.7% in those in their 50s, to 26.3% in those in their 70s, and to 20.8% in those aged 80 and above [7]. Clinically, FXTAS has a variable presentation with core features comprising cerebellar ataxia, intention tremor, parkinsonism, autonomic dysfunction, cognitive decline and psychological disorders [8]. Primary radiological markers are hyperintensive signals on T2-weighted MRI in specific brain regions, including the middle cerebellar peduncle (“MCP sign”), pons, corpus callosum and cerebral white matter, as well as generalized brain atrophy [9,10,11]. We recently expanded T2 findings in FXTAS to include abnormal signals in the globus pallidus (“pallidal sign”), a subcortical nucleus regulating movement, which displayed hyperintensities in the center surrounded by hypointensive T2 signals. We explored the clinical significance of both MCP and pallidal signs [12]. The prominence of the MCP sign in FXTAS pathophysiology was demonstrated by its independent associations with cerebellar ataxia, intention tremor and executive function deficits. Although the pallidal sign was not associated with motor or cognitive deficits independently, having both MCP and pallidal signs was associated with greater impairment in executive function and iron content variability in the globus pallidus [12].

Pathophysiologic mechanisms underlying the occurrence of T2 hyperintensities in FXTAS have not been explored. In elderly individuals with or without dementia, white matter hyperintensities (WMHs) on T2-weighted MRI are prevalent and thought to be associated with cognitive and motor deficits [13,14]. WMHs are commonly regarded as MRI features of cerebral small vessel disease (CSVD), supported by histopathological examinations of punctate and confluent deep WMHs that revealed small vessel pathologies, enlarged perivascular spaces, reactive gliosis and variable degrees of damage in axons, myelin and oligodendrocytes [15,16]. These findings are consistent with neuropathological changes in the white matter associated FXTAS, including gliosis and a variable degree of spongiosis, due to axonal and myelin loss [17]. Other MRI features of CSVD encompass small infarct, cerebral microbleed, enlarged perivascular spaces and brain atrophy [15,18]. Our recent neuropathological examination of microangiopathy [19] provided support for cerebrovascular dysfunction in FXTAS. An increased number of microbleeds in cerebral cortical white matter and cerebellum was discovered in FXTAS cases compared to age- and sex-matched control cases. Ubiquitin+ intranuclear inclusions, the pathological hallmarks of FXTAS, in the endothelial cells of capillaries were revealed as well, expanding the list of brain cell types that are compromised with the inclusions including neurons, astrocytes, ependymal cells, subependymal cells, the epithelial lining cells of the choroids plexus and Purkinje cells [17,20,21,22]. Neuropathologic examination of a male carrier with FXTAS revealed moderate CSVD in the arteries of deep white matter that exhibited wall thickening, perivascular gliosis and enlarged perivascular spaces and severe CSVD in the globus pallidus, showing calcification of the walls of perforating arteries [23]. Other prominent pathological features of FXTAS include RNA toxicity caused by elevated *FMR1* mRNA levels [24], iron accumulation in the putamen and choroid plexus [25,26] and mitochondrial dysfunction [27,28].

Cerebrovascular disease and its relationship with the appearance of T2-hyperintensities in FXTAS have not yet been investigated. Postmortem MRI has the advantages over in vivo MRI for examining brain changes at the end-stage of FXTAS, when patients are often bedridden. It also allows longer scanning time to provide improved image resolution, which is beneficial for detecting microbleeds with submillimeter diameters. In addition, postmortem MRI allows whole-brain detection and quantification of macroscopic pathological changes, including atrophy and MRI features of CSVD that are variably presented in patients with FXTAS and characterizes specific tissue properties, such as iron content. The findings can pinpoint specific brain regions for future histopathologic examinations at a microscopic scale. The goal of this study is to conduct a postmortem MRI study in FXTAS to examine (1) the severity of cerebrovascular disease characterized by intracranial hemorrhage, microbleeds, and WMHs; (2) the iron content in the white matter and 10 deep nuclei; (3) regional brain atrophy; and (4) the relationship between MRI measures and subgrouping among patients via principal component analysis (PCA). Our findings demonstrate that all cases with FXTAS showed hemorrhagic and/or ischemic brain damage, with T2-hyperintensities and iron depletion in the white matter, hippocampus and amygdala, indicating ischemic damage, and intracranial hemorrhage and iron accumulation in the basal ganglia, indicating hemorrhagic damage.

## 2. Materials and Methods

### 2.1. Sample Collection

Brain specimens were collected from 17 premutation carriers (males/females: 14/3) diagnosed of FXTAS during life and 7 non-carrier controls (males/females: 5/2) between 2009–2020 (Table 1). The diagnoses were confirmed in fixed brain tissue by the presence of intranuclear inclusions. The brains were procured from the Fragile X Brain Repository at the University of California Davis Medical Center, Sacramento, CA, USA. Written informed consent was obtained from the legal guardians/legally authorized representatives of the subjects for all specimens with the approval of the Institutional Review Board (IRB) of the University of California Davis Medical Center. All experiments were performed in accordance with relevant guidelines and regulations from the IRB of the University of California Davis Medical Center. All samples were fixed in 10% buffered formalin.

### 2.2. Postmortem MRI Acquisition

To remove background field effects and to keep the brain moist, formalin-fixed brains were placed in a plastic container filled with 3M fluorinert electronic liquid (FC-770, Parallax Technology, Inc., Waltham, MA, USA), which had a similar susceptibility as the brain [29]. The brains were rocked gently at room temperature for 12–24 h to allow air bubbles to escape [29,30,31]. MRI scans were acquired at a 3T Siemens Trio MRI scanner with a 32-channel head coil (Siemens Medical Solutions USA, Inc., Malvern, PA, USA) using 3D multi-echo gradient recalled echo (GRE) and 3D T2-weighted turbo spin echo sequences. For the initial 5 brains, multi-echo GRE scans were acquired in 112 slices of 0.5 mm thickness (no gap), with field of view = 224 mm^2^, matrix size = 448 × 448, repetition time = 50 ms, echo time (TE)_1_/spacing/TE_9_ = 5/5/45 ms, flip angle = 20° and number of excitations = 4. The initial 5 T2 scans were acquired in 224 slices of 0.5 mm thickness (no gap), with field of view = 256 mm^2^, matrix size = 512 × 512, repetition time = 3200 ms, TE = 371 ms, Turbo factor = 269 and number of excitations = 2. To minimize susceptibility artifacts caused by residual water/air bubbles, the subsequent 12 multi-echo GRE scans were acquired in 240 slices of 0.6 mm thickness (no gap), with field of view = 224 mm^2^, matrix size = 384 × 384, repetition time = 30 ms, TE_1_/spacing/TE_6_ = 2.25/2.25/13.50 ms, flip angle = 20° and number of excitations = 4. The subsequent 12 T2 scans were acquired in 288 slices of 0.5 mm thickness (no gap), with field of view = 256 mm^2^, matrix size = 512 × 512, repetition time = 3200 ms, TE = 371 ms, Turbo factor = 269 and number of excitations = 3.

### 2.3. Postmortem MRI Processing

Anterior and posterior commissures were aligned using DTI Studio (Laboratory of Brain Anatomical MRI and Center for Imaging Science at Johns Hopkins University, Baltimore, MD, USA) [32] and MRI bias field was corrected using N4 (https://github.com/ANTsX/ANTs/wiki/N4BiasFieldCorrection, accessed on 25 June 2023) [33]. A threshold was applied to the magnitude images of multi-echo GRE with the shortest TE to generate masks of the gray matter, white matter and ventricles. Anatomic regions comprising the cerebral gray and white matter, cerebellar gray and white matter, corpus callosum, cerebral peduncle, midbrain, pons and 10 subcortical nuclei (i.e., putamen, globus pallidus, caudate nucleus, thalamus, hippocampus, amygdala, subthalamic nucleus, red nucleus, substantia nigra and dentate nucleus) were segmented manually in ITK-Snap (http://www.itksnap.org/pmwiki/pmwiki.php, accessed on 25 June 2023) [34] using the generated gray matter/white matter masks, while referencing neuroanatomy books [35,36]. The segmentations were corrected iteratively until no errors or inconsistencies across the scans were found. The magnitude images were also used to estimate the R2* transverse relaxation rate by fitting a weighted-least-squares function on log-transformed signal intensities [37,38] using MATLAB R2014b (The MathWorks, Inc., Natick, MA, USA). The R2* transverse relaxation rate is affected by the degree of myelination and variations in iron concentration and, thus, can be used to detect demyelination and iron-containing blood degradation products due to intracranial hemorrhage or pathologic changes associated with small vessel disease such as microbleeds [37,39].

### 2.4. MRI Quantifications

Ratings of microbleeds and intracranial hemorrhages and estimations of R2* were performed for specific brain regions. Microbleeds were classified as well-defined, circular hyperintensities with diameters ranging from 2–10 mm on R2* map [39]. Only definite microbleeds in seven anatomic regions (i.e., frontal, parietal, temporal, occipital, insula, deep white matter and cerebellar regions) were counted and converted to the scale of 0–4, using cut-points (<1, 1–4, 5–9, 10–19, ≥20) [39,40]. The microbleeds rating of the cerebral cortex was then computed as the average rating of the frontal, parietal, temporal, occipital and insular regions. We also counted total brain regions showing intracranial hemorrhages [41,42] according to the following anatomical division: frontal, parietal, temporal, occipital and cerebellar regions. Rating of hyperintensities on T2-weighted scans followed the Fazekas method [43] in the following regions: anterior, posterior and inferior periventricular white matter; frontal, parietal, temporal and occipital deep white matter; MCP/cerebellar white matter; globus pallidus; brainstem; and the genu and splenium of the corpus callosum. Periventricular WMHs were rated as 0 (absence), 1 (“caps” or pencil-thin lining), 2 (smooth “halo”) and 3 (irregular hyperintensities extending into the deep white matter), whereas hyperintensities in the remaining brain regions were rated as 0 (absence), 1 (punctate foci), 2 (beginning confluence of foci) and 3 (large confluent areas). For the five whole brain specimens, the number of microbleeds and anatomic volume were estimated in both hemispheres and then divided by two. For hyperintensities, the higher ratings among the two hemispheres were utilized. Missing/incomplete brain regions were excluded from the analysis (Table 1). All ratings were performed after intra-rater reliability, assessed using Cohen’s kappa, reached 0.80 or above (almost perfect).

### 2.5. Histology

A sample of temporal cortex from P17M was dissected, rehydrated in 30% sucrose and embedded in optimal cutting temperature compound (Fisher HealthCare, Houston, TX, USA). Blocks were cut using a cryostat at 14 μm thickness. Tissue samples were processed according to standard procedures. Slides were rehydrated and stained with hematoxylin–eosin (H&E) dyes (Sigma-Aldrich SLCC6883/SLCJ2543, MilliporeSigma, Burlington, MA, USA), followed by dehydration in ethanol 50–100%, cleared in 3 changes of xylene, and mounted on a coverslip. Stained slides were imaged at 10× and 40× using a bright-field microscope (Olympus DP71, Evident Corporation, Tokyo, Japan).

### 2.6. Statistical Analysis

All statistical analyses were conducted using R 4.1.1 (R Foundation for Statistical Computing, Vienna, Austria, https://www.R-project.org/, accessed on 25 June 2023). The comparisons of MRI data between brains with FXTAS and control brains were conducted using multiple linear regression using age of death as a covariate. Because skulls were unavailable, we were unable to adjust individual differences in intracranial volume for volumetric measure. We substituted with sex to adjust intracranial volume differences between males and females. Multiple comparisons were corrected using the false discovery rate (FDR) [44]. Correlation among MRI measures and FXTAS subgroups were examined via PCA using the R package ‘factoextra’. PCA is a multivariate dimension-reduction statistical technique that performs linear transformation to represent a large set of correlated variables over a certain number of samples using a smaller set of uncorrelated principal components over the same samples. Principal components represent the underlying structure of the dataset, with highly correlated variables contributing to the same principal component, and each principal component explains a certain amount of variance within the dataset. Measurements included in the PCA analysis were age of death, T2-hyperintensity ratings, volumes and R2* transverse relaxation rate for the 10 nuclei, plus volumes of the cerebral and cerebellar gray and white matter, number of regions with intracranial hemorrhages and microbleed rating.

## 3. Results

### 3.1. T2-Hyperintensities

The average age of death for the premutation carriers (75.3 ± 8.0 years, range 66–93 years) was higher than that of non-carrier controls (70.1 ± 8.4 years, range 60–83 years). The MCP sign was observed in 1/7 control (C1M, cause of death: leukemia/respiratory failure) and in 12/17 FXTAS cases (male: 10/14, female: 2/3), while T2-hyperintensities in the globus pallidus was detected in 8 FXTAS cases only (male: 7/14, female 1/3). After adjusting for age of death, the ratings of MCP and pallidal hyperintensities were significantly higher in FXTAS cases than controls at FDR < 0.05 (MCP: *β* = 1.67 ± 0.62, FDR = 0.033; pallidus: *β* = 1.36 ± 0.48, FDR = 0.030), as well as the ratings in both the genu and splenium of the corpus callosum (genu: *β* = 0.96 ± 0.40, FDR = 0.041; splenium: *β* = 1.48 ± 0.39, FDR = 0.005). Confluent WMHs in the brainstem were detected in one female carrier and four male carriers but no controls while confluent WMHs in both genu and splenium of the corpus callosum were observed in four male carriers only. For the periventricular regions, 12/9/3 FXTAS cases showed confluent WMHs in the anterior/posterior/inferior regions; and for the deep white matter, WMHs reached confluency in 12/10/3/8 FXTAS cases in the frontal/parietal/temporal/occipital regions. Only one control (C4M) showed confluent WMHs in the parietal and occipital deep white matter. The ratings of the anterior periventricular region (*β* = 1.16 ± 0.28, FDR = 0.005) and frontoparietal deep white matter regions (frontal: *β* = 1.45 ± 0.39, FDR = 0.005; parietal: *β* = 1.05 ± 0.41, FDR = 0.035) were significantly higher in FXTAS cases than controls, after adjusting for age of death (Table 2).

### 3.2. Intracranial Hemorrhage and Microbleeds

Three FXTAS cases (P4M, P5M, and P7M) exhibited increased R2* consistent with intracranial hemorrhages affecting 2–3 brain regions (Figure 1a–c). One case, P17M, showed absence of the temporal white matter (Figure 1d), which could be caused by hemorrhages. H&E staining of the residual temporal cortex confirmed the presence of numerous small microbleeds with diameters in the order of micrometers (Figure 1e,f) and a large intracranial hemorrhage. None of the control cases showed intracranial hemorrhages. However, as a group, the FXTAS cases did not show significantly increased number of brain regions with intracranial hemorrhages compared with the controls after adjusting for age of death (β = 0.66 ± 0.43, *p* = 0.14) (Table 2). Ratings of microbleeds were not significantly different between the two groups in the cerebral cortex, deep white matter or cerebellum, after adjusting for age of death (β = −0.78 to 0.27, SE = 0.16 to 0.67, *p* = 0.06 to 0.62) (Table 2).

### 3.3. Anatomic Volume and R2* Transverse Relaxation Rate

We next compared anatomic volumes and R2* between the two groups by conducting multiple linear regression using age of death as a covariate. Sex was also used as a covariate for volumetric measures. Only the cerebellar white matter (*β* = −3.46 ± 1.09 cm^3^, FDR = 0.026), red nucleus (*β* = −0.08 ± 0.023 cm^3^, FDR = 0.023), and dentate nucleus (*β* = −0.37 ± 0.10 cm^3^, FDR = 0.021) demonstrated significant atrophy in the FXTAS cases (Table 3). The hippocampus (*β* = 3.43 ± 1.53, *p* = 0.036), subthalamic nucleus (*β* = 12.6 ± 4.60, *p* = 0.012) and substantia nigra (*β* = 9.22 ± 4.31, *p* = 0.045) showed increased R2* in the FXTAS group. However, the comparisons were not significant after controlling for the FDR (Table 4).

### 3.4. Principal Component Analysis (PCA)

Finally, PCA was performed to examine relationships among the MRI measures and heterogeneity among the 16 FXTAS cases. P17M was excluded because of missing data. Of the 16 identified PCA components, the first three components explained 29.1%, 25.8% and 11.1% of total variance, respectively (Appendix A). R2* (designated as d.~ in Figure 2a) of the cerebral white matter (d.wm) and cerebellar white matter (d.cbwm) and subcortical nuclei (e.g., d.thalamus) contributed the most to component 1, whereas anatomic volumes (designated as v.~ in Figure 2a) of the white matter (v.wm) and basal ganglia nuclei (e.g., v.pallidus) contributed the most to component 2. T2-hyperintensity ratings (designated as h.~ in Figure 2a) of the frontoparietal white matter (e.g., h.frontal) contributed to both components 1 and 2. In contrast, measurements of T2-hyperintensities of the globus pallidus (h.pallidus) and MCP, age of death (Age) and volumes of the gray matter (v.gm), red nucleus (v.red) and substantia nigra (v.nigra) were the main contributors to component 3 (Figure 2b). See Appendix A for contributions of variables in the determination of the 16 components and Appendix A for the quality of representations of the variables on the components (cos2). The qualities of representations of the 16 FXTAS cases are shown in Appendix A.

Figure 2 also shows negative relationships between some measures of iron content (indicated by “d.~” and having positive coordinates of component 1) and T2-hyperintensity ratings (“h.~” and having negative coordinates of component 1), both of which are orthogonal to anatomic volumes (“v.~” and having positive coordinates of component 2). Figure 2b shows the orthogonal relationships of T2-hyperintensities in the corpus callosum (h.genu and h.splenium), with both mean ratings of cerebral microbleeds (CMB) and total number of regions showing intracranial hemorrhages (ICH).

Plotting individual samples onto the PCA components in PCA biplots can identify clusters of cases with similar MRI measurements and MRI measurements separating different clusters of cases. The plots of individual FXTAS cases onto the PCA components in Figure 2 revealed clustering of the two female carriers who died at old age (93 years old for P2F and 89 years old for P3F) and showed low R2* in the subcortical nuclei (i.e., positioned in the opposite directions of the d.~ variables), while the third female that died at 79 (P14F) was positioned close to the center of gravity of the male FXTAS cases (the big green triangle), indicating similar MRI changes as the male cases (Figure 2a). The three male cases with intracranial hemorrhages (P4M, P5M, P7M) exhibited high coordinates of component 1 and showed relatively high R2* in the cerebral and cerebellar white matter and subcortical nuclei (Figure 2a). Males cases displaying relatively high volumes of the white matter and subcortical nuclei (top middle) also positioned away from those with relatively high ratings of WMHs in the frontoparietal regions (lower left) (Figure 2a).

Since the PCA revealed negative correlation between R2* in the cerebral and cerebellar white matter, hippocampus and amygdala (having positive values of component 1) and T2-hyperintensity ratings in the frontoparietal white matter (having negative values of component 1), we further explored their relationships by conducting multiple linear regression using age of death, group membership and individual T2-hyperintensity rating as the explanatory variables and R2* as the outcome variables. The results confirmed the negative correlations between R2* in these regions and frontoparietal T2-hyperintensity ratings. The analysis further indicated increased R2* in the cerebral white matter in FXTAS cases than controls after adjusting for frontal T2-hyperintensity ratings (*β* = 5.34–7.68, SE = 2.41–2.86, FDR = 0.022–0.046) and increased R2* in the hippocampus and amygdala in FXTAS cases after adjusting for posterior periventricular ratings (hippocampus: *β* = 4.48 ± 1.20, FDR = 0.009; amygdala: *β* = 3.24 ± 1.28, FDR = 0.028) (Table 5).

## 4. Discussion

We performed the first postmortem MRI study in FXTAS to quantify features of cerebrovascular disease and changes in R2* transverse relaxation rate in the white matter and subcortical nuclei that can be caused by demyelination, the presence of blood degradation products and iron accumulation in the subcortical nuclei. Correlations among different types of MRI measures and heterogeneity in patients were examined using the dimension reduction multivariate analysis technique, PCA.

All brains with FXTAS displayed MRI changes consistent with cerebrovascular disease. T2-hyperintensities in the white matter are commonly regarded as indications of CSVD [15,18]. All FXTAS cases exhibited multiple regions with confluent WMHs, except for two cases that showed a smooth “halo” (rating of 2) in the anterior periventricular white matter as the highest rating (Table 6). FXTAS cases showed significantly higher ratings of T2-hyperintensities in the MCP, globus pallidus, corpus callosum and frontoparietal white matter than controls, after the adjustment for age of death (Table 2). PCA revealed that high ratings of the frontoparietal WMHs were associated with low R2* decay in the cerebral white matter, implicating loss of oligodendrocytes, the predominant iron-containing cells in the brain [45], as well as ischemic, as opposed to hemorrhagic, injury that may underlie the occurrences of WMHs.

The cerebral white matter is more vulnerable to ischemia than the cortex due to the much lower artery and capillary density (2–3 times lower) [46]. Among the cerebral white matter regions, the frontal white matter is particularly susceptible to ischemia, where the blood is supplied by long and thin medullary arteries. This is in contrast with the subcortical U-fiber region, where shorter arteries provide the perfusion [15,46]. However, 4/17 FXTAS cases also showed increased R2* transverse relaxation rates in 1–3 cortical regions, in concordance with intracranial hemorrhages. Although ratings of microbleeds were not significantly higher in FXTAS cases than control cases, they were associated with a number of regions with intracranial hemorrhages (Figure 2b). These findings indicate that all brains with FXTAS show MRI changes consistent with ischemia, although hemorrhages could occur concurrently or even were the predominant injury in about 24% of the brains. Consistent with our findings, a recently published study [47] reported p62-positive intranuclear inclusions in the pericytes and endothelial cells of brain vasculature, as well as vascular infarcts such as lacunae and strokes throughout the brain and iron deposits resulting from disrupted vasculature throughout the cerebral cortex and hippocampus, in two male premutation carriers. These two men displayed mild motor impairments, no MCP sign or confluent WMHs, but prominent clinical symptoms of fragile X-associated neuropsychiatric disorders (FXAND), including apathy, aggression and depression [48]. In addition, the toxic polyglycine-containing protein, FMRpolyG protein [49], was also detected throughout the brain and brain vasculature [47], suggesting that compromised inclusion-bearing vasculature can be an important feature for both FXTAS and FXAND [19].

Brain regions showing significant atrophy in FXTAS cases were the cerebellar white matter, red nucleus and dentate nucleus (Table 3), among which atrophy of the red nucleus and dentate nucleus have not been reported in FXTAS. In vivo MRI studies [50,51] have demonstrated abnormal developmental trajectories of cerebellar and brainstem volumes in premutation carriers without FXTAS and reduced volumes in patients with FXTAS. We have also documented recently [12] higher iron concentrations in the dentate nucleus relative to controls and negative correlation between dentate volume and CGG repeat size in premutation carriers. The current study extended these findings by showing that the cerebellar white matter, dentate nucleus and red nucleus may be particularly vulnerable to FXTAS pathophysiology. The dentate nucleus, red nucleus and inferior olivary nucleus are interconnected by the central tegmental tract and the inferior and superior cerebellar peduncles, forming a triangular circuit important for learning and controlling fine voluntary movement [52,53,54]. Dysfunction of this triangular circuit may be critical for FXTAS symptomology. However, we were not able to adjust for individual differences in intracranial volume for volumetric measurements in the statistical analyses because the skull was unavailable. Also, in vivo MRI, commonly used for estimating intracranial volume, and head circumference (as a surrogate) were available only for 4–5 of the cases.

Unexpectedly, of the ten subcortical nuclei investigated in this study, only the hippocampus, subthalamic nucleus and substantia nigra suggested a higher R2* in FXTAS cases, compared to controls, that were not significant after the correction for multiple testing. However, after adjusting for frontoparietal WMH ratings, the cerebral and cerebellar white matter, hippocampus and amygdala revealed higher R2* in FXTAS cases than controls (Table 5). R2* in the human brain is affected by diamagnetic myelin, paramagnetic iron and blood degradation products, and orientation of myelinated axons relative to external magnetic field [39,55]. Hence, the relatively higher R2* in the white matter in FXTAS may be due to demyelination and/or increased iron content carried by surviving oligodendrocytes [45]. This is consistent with neuropathologic findings of white matter spongiosis with corresponding axonal loss and myelin pallor in FXTAS [17]. In contrast, the relatively higher R2* in subcortical nuclei (with low myelin content) can be caused by hemosiderin depositions that have been demonstrated histologically in parenchyma and capillaries of the putamen [26]. We were not able to replicate the finding of high iron content in the dentate nucleus in premutation carriers from our recent in vivo MRI study [12]. This may be due to changes in iron content at different stages of FXTAS, since we showed that iron content decreased as the dentate nucleus atrophied [12]. Further MRI-pathologic association studies are needed to clarify the source of increased R2* in the white matter and deep nuclei.

One strength of this study was the characterization of heterogeneity in FXTAS via PCA, which revealed subgroups that varied by sex, age of death, iron content in the subcortical nuclei, severity of frontoparietal WMHs and degree of atrophy in the cerebral and cerebellar white matter and subcortical nuclei. This can be helpful for recognizing the range of MRI changes associated with FXTAS and for developing effective personalized therapeutic treatments to alleviate or reverse these changes. However, we were not able to explore clinical significance of the subgrouping, since clinical data were unavailable from many cases with FXTAS. In addition, quantitative susceptibility mapping was conducted following our published method [12] but was not usable because of substantial artifacts from residual air and/or water bubbles.

In conclusion, we revealed MRI changes in the brain consistent with cerebrovascular disease in all 17 cases of FXTAS. All FXTAS cases exhibited WMHs that were associated with reduced R2*, indicating the loss of iron-containing oligodendrocytes and ischemic damage in the white matter. Four FXTAS cases (23.5%) also showed increased R2* in 1–3 brain regions, consistent with intracranial hemorrhages. Those with intracranial hemorrhages tended to show increased R2* in the basal ganglia supporting hemorrhagic nature of both types of changes. Recognizing cerebrovascular disease as an important feature of FXTAS may prove to be beneficial for discovering effective FXTAS prevention and treatment.

## Figures and Tables

**Figure 1 cells-12-01898-f001:**
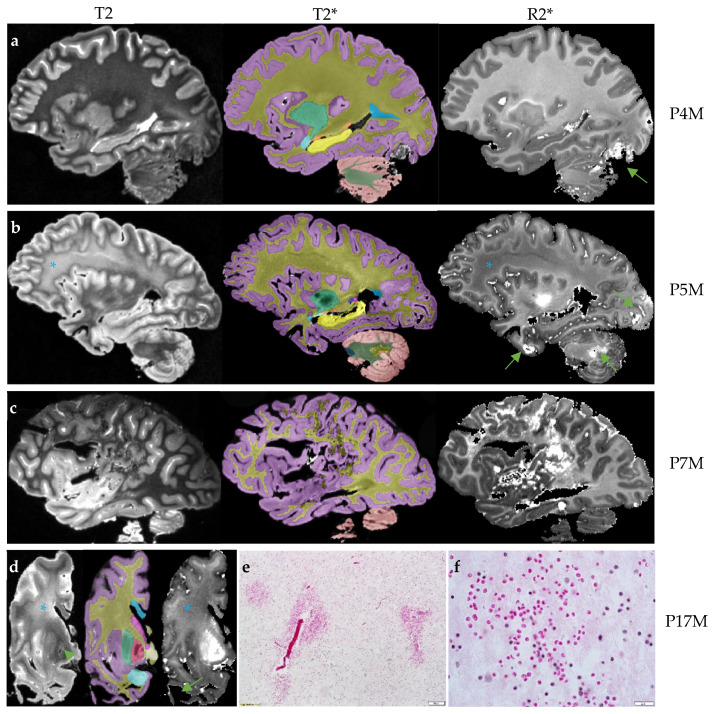
The four brains with intracranial hemorrhages. The T2 scan for rating hyperintensities, T2* scan for anatomic segmentation and R2* mapping for rating cerebral microbleeds and estimating R2* transverse relaxation rates are shown for each brain. (**a**) The R2* mapping of P4M shows an intracranial hemorrhage affecting the occipital lobe and cerebellum (arrow). The T2 scan shows confluent T2-white matter hyperintensities in the adjacent regions while other white matter regions appear to be normal. (**b**) The R2* mapping of P5M shows intracranial hemorrhages affecting the occipital lobe, anterior temporal lobe and cerebellum (arrows). The frontoparietal white matter shows confluent white matter hyperintensities on the T2 scans and reduced R2* transverse relaxation rate on the R2* mapping (*). (**c**) The R2* mapping of P7M shows a large intracranial hemorrhage affecting the frontal, parietal and occipital lobes. (**d**) The scans of P17M show confluent frontal white matter hyperintensities on the T2 scan and loss of iron content on R2* mapping in the same region (*). P17M also shows T2-hyperintensities in the globus pallidus (arrow) and corresponding signal variations on the R2* mapping. All T2, T2* and R2* mapping show the absence of anterior temporal white matter (arrow). (**e**,**f**) The hematoxylin and eosin (H&E) staining of the residual temporal cortex of P17M shows numerous free erythrocytes, indications of microbleeds. Scale bar: 100 µm in E and 20 µm in F. On T2* scans, the cerebral cortex is labeled as lilac; cerebral white matter, dark yellow; corpus callosum, blue; caudate nucleus, pink; tail of caudate nucleus, purple; putamen, green; globus pallidus, red; amygdala, light blue; hippocampus, yellow; unnamed subcortical gray matter, vanilla; pons, dark blue; cerebellar cortex, salmon; cerebellar white matter, dark green; and cerebellar dentate nucleus, sand.

**Figure 2 cells-12-01898-f002:**
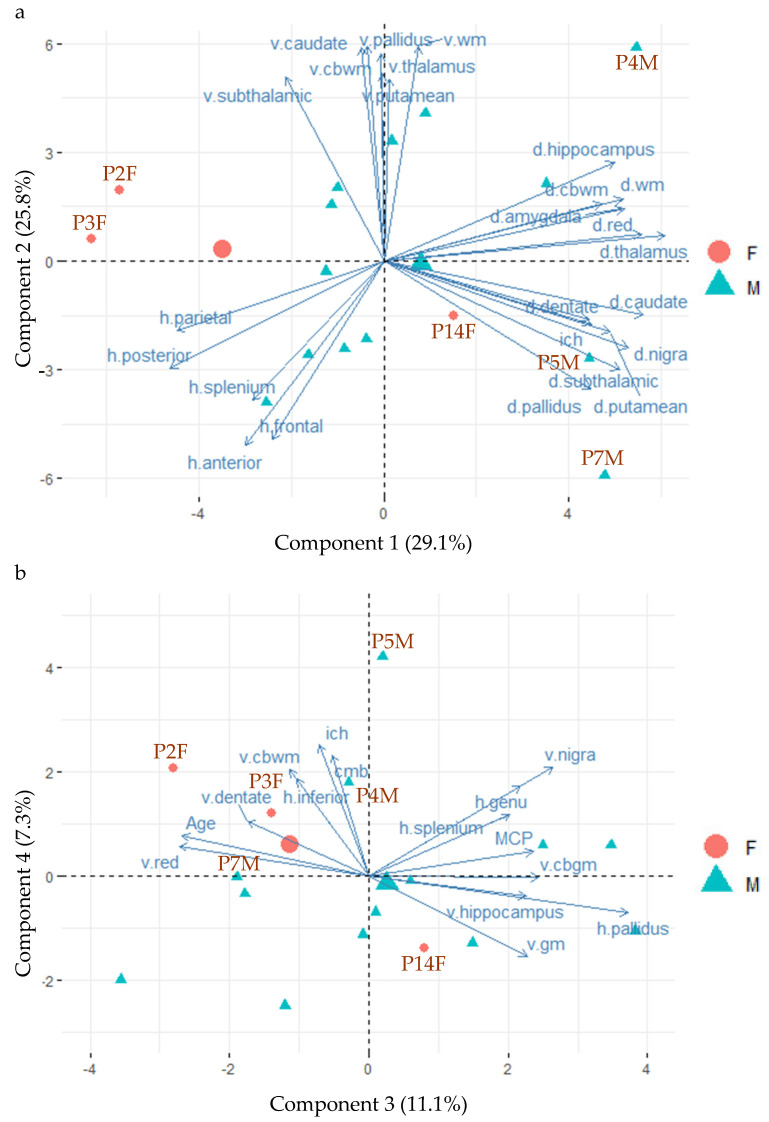
The biplots of principal component analysis showing the positions of 16 FXTAS case (P17M is excluded because of missing data) and the main contributing MRI measures on the first four components. (**a**) The positions of the 16 FXTAS cases and the top 25 contributing MRI measures for components 1 and 2. (**b**) The positions of the 16 FXTAS cases and the top 15 contributing MRI measures for components 3 and 4. Abbreviations: d.~ = iron content; h.~ = T2 hyperintensities; v.~ = volume; anterior = anterior periventricular white matter; cbwm = cerebellar white matter; cmb = cerebral microbleed; gm = gray matter; ich = intracranial hemorrhage; posterior = posterior periventricular white matter; MCP = the middle cerebellar peduncle sign; n. = nucleus; anterior = anterior periventricular white matter; wm = white matter.

**Table 1 cells-12-01898-t001:** Characteristics of the brains with fragile X-associated tremor/ataxia syndrome (FXTAS) and of controls.

ID	Age	Sex	PMI (Hour)	Brain Type	CGG	Cause of Death	Mid-Brain *	Pons *	CB *
P1M	68	M	84	WB	119	FXTAS-related complications	1	1	1
P2F	93	F	NA	LH	60, 30	FXTAS-related complications	1	1	1
P3F	89	F	48	RH	71, 30	FXTAS-related complications	1	1	1
P4M	75	M	6–8	LH	67	FXTAS-related complications	1	1	1
P5M	71	M	12	RH	120	FXTAS-related complications	1	1	1
P6M	67	M	NA	WB	NA	FXTAS-related complications	1	1	1
P7M	69	M	18	RH	118	FXTAS-related complications	1	0	1
P8M	77	M	5.3	RH	95	FXTAS-related complications	1	1	1
P9M	66	M	NA	RH	93	FXTAS-related complications	0	0	1
P10M	82	M	29	RH	70	FXTAS-related complications	0	0	1
P11M	85	M	NA	RH	66	FXTAS-related complications	0	0	1
P12M	72	M	3	LH	60	FXTAS-related complications	1	1	1
P13M	74	M	NA	WB	NA	FXTAS-related complications	1	1	1
P14F	79	F	NA	LH	78, 30	FXTAS-related complications	0	0	1
P15M	71	M	NA	WB	76	FXTAS-related complications	1	1	1
P16M	70	M	21	WB	NA	FXTAS-related complications	1	1	1
P17M	71	M	19	RH	85	FXTAS-related complications	0	0	0
C1M	65	M	42.2	LH	NA	Leukemia/respiratory failure	1	1	1
C2M	74	M	136.1	LH	NA	Cardiovascular disease	1	1	1
C3M	62	M	36.8	LH	NA	Cardiopulmonary arrest	1	1	1
C4M	77	M	78.0	LH	NA	Cancer	1	1	1
C5F	60	F	113.5	LH	NA	Cirrhosis, alcohol use disorder	1	1	1
C6M	83	M	208	LH	NA	Cancer	1	0	1
C7F	70	F	55	LH	NA	Unknown	0	0	1

* 1 = complete presence of the tissue; 0 = absence or incomplete presence of the tissue. Abbreviations: CB = cerebellum; LH = left hemisphere; RH = right hemisphere; WB = whole brain; PMI = postmortem interval. The identifications (IDs) for all FXTAS cases start with “P” while those of control cases start with “C”.

**Table 2 cells-12-01898-t002:** Comparisons of MRI features of cerebrovascular disease between brains with FXTAS and control brains.

Brain Regions	FXTAS			Control			Group Comparisons			
	*N*	Mean	SD	*N*	Mean	SD	*β*	SE	*p* Value	FDR
Subcortical T2-hyperintensities										
MCP	17	1.88	1.41	7	0.43	1.13	1.67	0.62	0.014	**0.033**
Globus pallidus	17	1.06	1.30	7	0	0	1.36	0.48	0.010	**0.030**
Brainstem	12	2.08	0.90	6	1.33	0.52	0.76	0.43	0.10	0.15
CC genu	17	1.94	0.90	7	1.00	0.58	0.96	0.40	0.024	**0.041**
CC splenium	17	1.94	0.90	7	0.43	0.54	1.48	0.39	0.001	**0.005**
PV: anterior	17	2.65	0.61	7	1.43	0.54	1.16	0.28	0.0004	**0.005**
PV: posterior	17	2.18	1.02	7	1.57	0.79	0.46	0.45	0.31	0.34
PV: inferior	17	1.35	0.93	7	1.29	0.49	−0.12	0.38	0.76	0.76
DWM: frontal	17	2.41	0.94	7	0.86	0.38	1.45	0.39	0.001	**0.005**
DWM: parietal	17	2.24	0.97	7	1.14	0.38	1.05	0.41	0.018	**0.035**
DWM: temporal	17	1.29	0.99	7	0.86	0.38	0.60	0.39	0.14	0.19
DWM: occipital	17	2.00	1.00	7	1.29	0.76	0.57	0.44	0.21	0.25
Microbleeds										
Cerebral cortex	17	1.37	0.74	7	2.11	1.01	−0.78	0.39	0.06	-
DWM	17	2.18	1.24	7	2.29	1.89	−0.34	0.67	0.62	-
Cerebellum	17	3.00	0.00	7	2.67	0.58	0.27	0.16	0.10	-
Intracranial hemorrhage										
# of regions	17	0.47	0.94	7	0	0	0.66	0.43	0.14	-

Bold, FDR < 0.05. Abbreviations: CC = corpus callosum; DWM = deep white matter; FDR = false discovery rate; FXTAS = fragile X-associated tremor/ataxia syndrome; MCP = middle cerebellar peduncle; PV = periventricular; SD = standard deviation; SE = standard error.

**Table 3 cells-12-01898-t003:** Comparisons of anatomic volumes between brains with FXTAS and control brains.

Brain Regions	FXTAS			Control			Group Comparisons			
	*N*	Mean	SD	*N*	Mean	SD	*β*	SE	*p* Value	FDR
Cerebral WM	17	141.8	32.4	7	172.6	42.0	−33.4	17.4	0.07	0.10
Cerebral GM	17	261.8	30.3	7	287.2	44.2	−26.5	16.6	0.13	0.17
Cerebellar WM	16	6.81	1.94	7	10.05	2.59	−3.46	1.09	0.005	**0.026**
Cerebellar GM	16	42.80	7.39	7	52.93	8.14	−8.55	3.73	0.034	0.10
Corpus callosum	17	8.12	2.82	7	9.55	2.56	−1.67	1.33	0.23	0.28
Cerebral peduncle	16	0.68	0.21	7	0.90	0.20	−0.20	0.10	0.07	0.10
Putamen	17	4.44	0.50	7	4.89	0.71	−0.53	0.27	0.07	0.10
Globus pallidus	17	1.42	0.28	7	1.64	0.26	−0.26	0.13	0.07	0.10
Caudate N.	17	3.74	0.63	7	3.96	0.40	−0.26	0.29	0.39	0.45
Thalamus	17	6.02	1.10	7	7.42	1.32	−1.39	0.57	0.024	0.09
Hippocampus	17	3.41	0.62	7	4.14	0.87	−0.61	0.31	0.07	0.10
Amygdala	17	1.47	0.39	7	1.68	0.42	−0.10	0.18	0.59	0.62
Subthalamic N.	17	0.08	0.04	7	0.07	0.02	0.008	0.017	0.62	0.62
Red N.	16	0.12	0.06	7	0.18	0.04	−0.08	0.023	0.003	**0.023**
Substantia nigra	16	0.47	0.14	7	0.63	0.12	−0.15	0.07	0.036	0.10
Dentate N.	17	0.79	0.24	7	1.10	0.16	−0.37	0.10	0.001	**0.021**

Bold, FDR < 0.05. Abbreviations: FDR = false discovery rate; FXTAS = fragile X-associated tremor/ataxia syndrome; GM = gray matter; MCP = middle cerebellar peduncle; N. = nucleus; PV = periventricular; SD = standard deviation; SE = standard error; WM = white matter.

**Table 4 cells-12-01898-t004:** Comparisons of iron content between brains with FXTAS and control brains.

Brain Regions	FXTAS			Control			Group Comparisons			
	*N*	Mean	SD	*N*	Mean	SD	*β*	SE	*p* Value	FDR
Putamen	17	36.4	8.3	7	41.5	9.3	−2.92	3.82	0.45	0.68
Globus pallidus	17	59.0	20.4	7	55.3	8.4	9.43	7.41	0.22	0.39
Caudate N.	17	30.8	7.3	7	29.0	4.2	3.57	2.89	0.23	0.39
Thalamus	17	31.1	4.5	7	29.6	3.8	2.98	1.75	0.10	0.31
Hippocampus	17	24.9	3.7	7	22.3	2.7	3.43	1.53	0.036	0.18
Amygdala	17	23.2	3.8	7	22.0	2.9	2.27	1.54	0.15	0.37
Subthalamic N.	17	52.8	13.0	7	44.1	5.7	12.6	4.60	0.012	0.15
Red N.	16	53.1	8.6	7	55.8	10.4	0.24	3.84	0.95	0.95
Substantia nigra	16	52.5	12.0	7	47.1	6.0	9.22	4.31	0.045	0.18
Dentate N.	17	41.4	7.9	7	45.0	9.9	−1.51	3.74	0.69	0.83
Cerebral WM	17	35.7	6.2	7	35.3	3.1	1.42	2.56	0.59	0.78
Cerebellar WM	17	33.6	5.2	7	33.6	2.6	0.55	2.11	0.80	0.87

Abbreviations: FXTAS = fragile X-associated tremor/ataxia syndrome, FDR = false discovery rate; SD = standard deviation; SE = standard error; WM = white matter.

**Table 5 cells-12-01898-t005:** Group comparisons of iron content after adjusting for age of death and T2-hyperintensity rating.

R2* Transverse Relaxation Rate	Group Comparisons				T2-Hyperintensity Rating as a Covariate				
	*β*	SE	*p* Value	FDR	Region	*β*	SE	*p* Value	FDR
Cerebral WM	7.68	2.86	0.014	**0.022**	PV anterior	−5.42	1.66	0.004	**0.011**
Cerebral WM	2.81	2.28	0.23	0.23	PV posterior	−3.03	1.09	0.011	**0.019**
Cerebral WM	6.59	2.84	0.031	**0.039**	DWM frontal	−3.56	1.24	0.009	**0.019**
Cerebral WM	5.34	2.41	0.039	**0.046**	DWM parietal	−3.75	1.13	0.003	**0.011**
Cerebellar WM	4.98	2.37	0.049	0.053	DWM frontal	−3.06	1.05	0.009	**0.019**
Hippocampus	4.48	1.20	0.0013	**0.009**	PV posterior	−2.27	0.57	0.001	**0.009**
Amygdala	3.24	1.28	0.020	**0.028**	PV posterior	−2.10	0.61	0.003	**0.011**

BOLD, significant at FDR 0.05. Abbreviations: DWM = deep white matter; PV = periventricular; SE = standard error; WM = white matter.

**Table 6 cells-12-01898-t006:** Brain regions with confluent T2-hyperintensities (as indications of ischemic damage) and intracranial hemorrhages for the 17 cases with FXTAS.

ID	Age	Regions with Confluent T2-Hyperintensities	Regions with Intracranial Hemorrhage	Brain Damage Type
P1M	68	GP, aPV, fDW, pDW, oDW	NA	Ischemic
P2F	93	Brainstem, aPV, iPV, pPV, fDW, pDW, oDW	NA	Ischemic
P3F	89	aPV, iPV, pPV, fDW, pDW, oDW	NA	Ischemic
P4M	75	MCP, brainstem, oDW	Occipital and cerebellum	Both
P5M	71	MCP, CCg, aPV, fDW	Temporal, occipital, and cerebellum	Both
P6M	67	MCP, GP, brainstem	NA	Ischemic
P7M	69	aPV, fDW, tDW	Frontal, parietal, and temporal	Both
P8M	77	MCP, brainstem, aPV, pPV, fDW, pDW, oDW	NA	Ischemic
P9M	66	MCP, aPV, pPV, fDW, pDW, tDW	NA	Ischemic
P10M	82	aPV (2) *	NA	Ischemic
P11M	85	MCP, CCg, CCs, aPV, pPV, fDW, pDW	NA	Ischemic
P12M	72	aPV (2) *	NA	Ischemic
P13M	74	MCP, GP, brainstem, CCg, CCs, aPV, pPV, fDW, pDW, oDW	NA	Ischemic
P14F	79	MCP, aPV, fDW	NA	Ischemic
P15M	71	pPV, pDW, oDW	NA	Ischemic
P16M	70	MCP, CCg, CCs, aPV, pPV, fDW, pDW	NA	Ischemic
P17M	71	MCP, GP, CCg, CCs, aPV, iPV, pPV, fDW, pDW, tDW, oDW	Temporal	Both

* The highest rating of the white matter hyperintensities was 2 in the anterior periventricular white matter (smooth “halo”). Abbreviations: CCg = genu of the corpus callosum, CCs = splenium of the corpus callosum, fDW = frontal deep white matter, oDW = occipital deep white matter, pDW = parietal deep white matter, tDW = temporal deep white matter, GP = globus pallidus, MCP = middle cerebellar peduncle, aPV = anterior periventricular white matter, iPV = inferior periventricular white matter, pPV = posterior periventricular white matter.

## Data Availability

The data presented in this study are available on reasonable request from the corresponding authors.

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
