# Peer review of "A Postmortem MRI Study of Cerebrovascular Disease and Iron Content at End-Stage of Fragile X-Associated Tremor/Ataxia Syndrome"

_cells, 2023, doi:10.3390/cells12141898_

Round 1

Reviewer 1 Report

Review:

This postmortem MRI study of individuals with fragile X-associated tremor/ataxia syndrome (FXTAS) quantified cerebrovascular disease, brain atrophy, and iron content and revealing the presence of intracranial hemorrhage, T2-hyperintensities, and significant atrophy in specific brain regions. Such findings may have some clinical implications for the fragile X-associated tremor/ataxia disease given that this study is based on postmortem MRI and is the first study to date.

I'm primarily interested in statistical analysis. I've provided some feedback below.

1)    It is unclear which covariates were controlled in each analysis. It appears to me that the authors adjusted the effect of age of death in all group comparisons when they wrote, "The comparisons of MRI data between brains with FXTAS and control brains …. using age of death as a covariate". However, based on the statements in Results, it seems like they only controlled covariates in Section 3.3 and 3.4 but not 3.2 or 3.1.

2)    I'm curious as to why the authors only include death age as a covariate. Did the writers look into how education and age might have an effect on those brain features? Additionally, I'd advise the author to take the effect of intracranial volume (ICV) into account when looking into the subcortical volumes.

 3) In the method section, the authors should include additional information about PCA analysis. For example, if they have pre-defined the number of components in the PCA analysis, and if so, how many did they suggest. Additionally, I'd advise the authors to include the full PCA results in a separate table. Taking each input variable's contribution to each component as an example.

4) It took me some time to understand Figure2. I'd want to add a couple comments to that. I would advise the authors to: i) use as few abbreviations as possible to represent the results in the figure and figure legend; ii) please clarify any ambiguities in your figure legend in a way, such as the size of the triangle; and iii) certain information in the existing figure legend should be moved to Results.

Minor comments:

1) Some text in the Result section should be moved to Method, "PCA is a multivariate dimension-reduction‚...and microbleed rating." (Lines #235-244).

2) Line #46: it'd be great if the authors report the prevalence of FXTAS across sexes rather than only in males.

Overall, the quality of English language looks ok to me. But It'd be great if the authors could make the draft more layman-friendly. Also, some content should be relocated (see comments above).

Reviewer 2 Report

In the present study, the authors performed a MRI postmortem on brains from FXS patients and age-matched controls in order to assess iron content and possible brain atrophy and cerebrovascular alterations. The use of PCA on data has the advantage to dissect and valuate the specific role of each of the above parameters, as well as their possible relations. This novel approach allows the authors to identify, among others, iron depletion as a marker for ischemic damage and iron accumulation to label hemorrhagic damage in the FXS brains. Furthermore, the evidence of cerebrovascular alterations in FXS brains as assessed from this post-mortem study may provide a valid background for future studies using live imaging on performing individuals (i.e. fMRI studies). Overall, the manuscript is clear and the study is well-conducted. I only have one minor concern: although novel, the use of MRI on post-mortem tissues is quite unusual. Authors should better declare the rationale for this approach. Also, literature supporting the study's driving hypothesis can be more detailed. 

Round 2

Reviewer 1 Report

Overall I'm happy with the authors' revisions. However, I'd add a minor suggestion to the authors for figure. It'd be great if the authors consider reducing the complexity of the figure by simplifying the findings or removing any unnecessary details or elements that may distract from the main findings. 

Looks fine to me.